# Developing Low-Cost Mobile Device and Apps for Accurate Skin Spectrum Measurement via Low-Cost Spectrum Sensors and Deep Neural Network Technology

**DOI:** 10.3390/s22228844

**Published:** 2022-11-15

**Authors:** Ling-Cheng Hsu, Shiang Hsu, Tan-Hsu Tan, Chia-Hsing Cheng, Cheng-Chun Chang

**Affiliations:** 1Department of Electrical Engineering, National Taipei University of Technology, Taipei 10608, Taiwan; 2Department of Electrical Engineering, National Formosa University, Yunlin 632301, Taiwan

**Keywords:** low-cost miniature skin spectrum measurement system, deep neural network, miniature spectrum sensing chip

## Abstract

In recent years, skin spectral information has been gradually applied in various fields, such as the cosmetics industry and clinical medicine. However, the high price and the huge size of the skin spectrum measurement device make the related applications of the skin spectrum unable to be widely used in practical applications. We used convolutional neural network (CNN) to achieve a satisfying accuracy of the Fitzpatrick skin-type classification by using a simple self-developed device in 2018. Leveraging on the hardware, firmware, and software app-developing experience, a low-cost miniature skin spectrum measurement system (LMSSMS) using deep neural network (DNN) technology was further studied, and the feasibility of the system is verified in this paper. The developed LMSSMS is divided into three parts: (1) miniature skin spectrum measurement device (MSSMD), (2) DNN model, and (3) mobile app. The MSSMD was developed with innovative low-cost MSSC, 3D printing, and a simple LED light source. The DNN model is designed to enhance measurement accuracy. Finally, the mobile app is used to control and show the measurement results. The developed app also includes a variety of skin-spectrum-related applications, such as erythema index and melanin index (EI/MI) measurement, Fitzpatrick skin-type classification, Pantone SkinTone classification, sun-exposure estimation, and body-fat measurement. In order to verify the feasibility of LMSSMS, we used the standard instrumentation device as a reference. The results show that the accuracy of the LMSSMS can reach 94.7%, which also confirms that this development idea has much potential for further development.

## 1. Introduction

Skin spectrometry has many applications worth exploring in the fields of medicine, cosmetology, and personal healthcare. In medicine, it can be used for preliminary physical health testing [1,2]. For example, liver and gallbladder disease may cause yellowing of the skin [3]; cardiovascular diseases such as blocked blood vessels can cause pallor, cyanosis, and yellowing [4]; and some allergic reactions lead to a red rash or erythema on the skin [5]. The change of skin tone can be accurately obtained by measuring the skin spectrum. Therefore, the skin reflection spectrum can be regarded as one of the important reference indicators of physical health. In cosmetology, the skin spectrum is an important indicator to assist the selection of cosmetics and skincare products. For example, the erythema index and melanin index (EI/MI) can be used to measure skin-tone changes, the Fitzpatrick skin-type classification is often used to judge the degree of skin pigmentation, and the comparison between skin tone and Pantone SkinTone Guide is used for the selection of base cosmetics products. In personal healthcare, it is possible to calculate the appropriate daily sun-exposure time by measuring the skin tone. In this way, this allows the body to synthesize vitamin D, and at the same time protects humans against sunburn, and also provides an estimate of body fat percentage.

However, currently available skin tone or skin spectrum measuring devices present different problems for individual daily use. Although professional instruments such as CM-700d/CM-600d (Konica Minolta, Inc., Tokyo, Japan) [6] and Mexameter^®^ Mx 18 (Courage + Khazaka electronic GmbH, Cologne, Germany) [7] can serve as golden standard, they are bulky, heavy, and expensive. Apps that use smartphone’s built-in cameras, such as Microskin™ ColourMatch (Kerim Health Ltd., Maidenhead, UK) [8] and My Best Colors (colorwise.me) [9] are easy to use, but are likely to be affected by many factors such as the camera quality, shooting environments, and white balance calibration, leading to inaccurate results. On the other hand, hand-held skin tone measurement devices such as PANTONE CAPSURE RM200-PT01 (X-Rite, Inc., Grand Rapids, MI, USA) [10] and SKINCOLORCATCH (Delfin, Kuopio, Finland) [11] have emerged on the market and showed a great potential for daily use applications, due to a reasonable price, accuracy and easy usage.

It is worth noting that in recent years, with the advancement of semiconductor manufacturing processes and micro electromechanical systems (MEMS) technology, new miniature spectrum sensing chips (MSSCs) have emerged increasingly, such as C12666MA [12], NSP32 [13] and so on. These new types of spectrum sensing wafers solve the shortcomings of traditional optical spectrometers with bulky and fragile optical components. The low cost and small size of the MSSCs make them easy to be integrated into other devices. However, the measurement accuracy of skin spectrum will be affected by the factors such as the sensitivity and resolution of the MSSCs, the optical path and the lighting source; however, there was still room for improvement in the measurement accuracy of the spectrum. Supervised deep learning has been highly successful on classification in recent years, achieving state-of-the-art results in many tasks. We conducted Fitzpatrick skin type classification with a self-developed simple device using convolutional neural network (CNN) in 2018 [14], achieving a 92.59% classification accuracy. Recently, deep learning classification has been developed in the era of self-annotation, active sampling, and self-learning, making the classification tasks even more intelligent [15]. Apart from classification, this study is trying on super resolution capability of deep learning. For proof-of-concept development, this study adopted the well-developed VDSR for spectral enhancement. Leveraging on the developing experience in [14], this study developed a low-cost miniature skin spectrum measurement system (LMSSMS) using deep neural network (DNN) technology. The system includes (1) a miniature skin spectrometry device developed by using low-cost MSSCs, 3D printing device housing, and a simple LED light source; (2) a well-established deep neural network technique for enhancing measurement accuracy; and (3) a mobile app to control and display the measurement results. In addition, a variety of related applications that rely on skin spectrum measurement are integrated into the mobile app, including EI/MI measurement, Fitzpatrick type classification, Pantone SkinTone classification, sun-exposure estimation, and body-fat measurement.

The organizational structure of this paper is as follows. Section 2 presents the design of the LMSSMS. Section 3 shows the system verification experiments and results. Section 4 is the conclusion.

## 2. LMSSMS Design

The structure of the LMSSMS is shown in Figure 1, which is divided into three parts: (1) miniature skin spectrum measurement device (MSSMD), (2) DNN model, and (3) mobile app.

### 2.1. MSSMD

The MSSMD is mainly divided into four parts for design and development: (1) miniature spectrum measurement element, (2) measurement light source, (3) optical mechanism, and (4) control and communication interface. They are respectively explained as follows:

#### 2.1.1. Miniature Spectrum Measurement Element

The traditional spectrum measurement element needs to complete the dispersion by a large volume and precise optical mechanism for the measurement. Although the measurement is more accurate, it is difficult to reduce the cost and achieve miniaturization. Therefore, this study focused on using a novel MSSC NSP32 developed by nanoLambda as a chip for skin spectrometry. The NSP32-W1 spectrum chip uses a 1024-pixel, 32 × 32 nano-optical filter array to measure the light spectrum, based on a technological breakthrough, the plasmonic filter. The 100 μm-high sensor die is assembled in an advanced ball grid array (BGA) package with an optical window for only 500 μm height. The other optical parts, the lens and the diffuser, are assembled in a 5.7 mm–high module. The module is very small overall, at just 6 mm × 6 mm × 5.3 mm. Spectrum measurement range is wide from 400 nm to 1000 nm. Therefore, it is very suitable for the use in developing miniature spectrum measurement systems.

#### 2.1.2. Measurement Light Source

The skin spectrum is the spectrum of light reflected from the light impinging on the skin. Therefore, a full wavelength light source is required to illuminate the skin during measurement. LEDs are the best choice for miniature and low-cost light sources. In this study, two SunLike LEDs and two NIR LEDs developed by Seoul Semiconductor were selected as measurement light sources. The specifications and configurations are shown in Figure 2. The light source emitted by SunLike LED STW9C2PB-S covers the entire visible light, and the light intensity of 400~800nm in the spectrum is quite an average [16]. This enables more accurate skin-reflectance spectrum measurements. In addition, two additional sets of near-infrared light sources—SFH-4786S [17], with a wavelength centered at 810 nm, and VSMY14940 [18], with a wavelength centered at 940 nm—were added. The overall light source can cover the measurement range of the NSP32.

#### 2.1.3. Optical Mechanism

The design of the optical mechanism will affect the optical path during measurement, and this, in turn, affects the accuracy of the spectrum measurement results. In order to reduce the cost, the 3D design software Sketch Up 2018 was used in this study to design a simple, low-cost, available optical path and compact optical mechanism, as shown in Figure 3. The light source is arranged around the NSP32 to illuminate the skin, and the NSP32 in the middle measures the reflection spectrum of the skin. In addition, a ring-shaped wall was designed between the NSP32 and the light source as a light shield to reduce the direct interference of light leakage and external light sources to the NSP32.

#### 2.1.4. Control and Communication Interface

The miniature skin spectrometry device has a very compact design. In order to control the device for skin spectrum measurement, process the measured data, receive the measurement command, and return the measurement result, we designed the control circuit and communication interface of the MSSMD, using the microcontroller STM32F411 and the Bluetooth chip nRF52832. The STM32F411 is mainly responsible for processing the spectral-measurement-related commands received by nRF52832, controlling the light source switch, capturing NSP32 spectral signals, and returning the measured spectral data to nRF52832. The nRF52832 is mainly responsible for receiving measurement commands and returning measurement results, using Bluetooth.

Finally, the power supply of the system adopts USB power supply and lithium battery. The device housing is designed according to the internal space requirements, such as the volume of individual components. The final size is set to 38.5 mm × 36 mm × 30 mm. The finished device is shown in Figure 4a–c. In order to test the spectrum of the module light source, we place the developed skin-reflectance spectrum measurement module on a standard optical diffuse reflectance whiteboard WS-1-SL. The measured result from the developed device is shown in Figure 5. This result serves as the baseline for calibration of the skin spectral-measurement module.

### 2.2. DNN Model

Due to the simplicity and low cost of MSSMD design and components, spectral signal accuracy still needs to be improved. This study uses the DNN technique of VDSR [19] to enhance the measurement accuracy. However, the input and output data of VDSR are two-dimensional (2D), while the spectral signal is one-dimensional (1D). Therefore, a conversion program is required to convert between input/output data and spectral data.

The VDSR architecture parameters used in this study are set as follows: The number of convolutional layers is 20. The first layer operates on the input image. The last layer of the convolutional layer is used to reconstruct the image with only 1 filter. Each of the remaining convolutional layers has 64 filters. The kernel size is 3 × 3. The size of the receptive field is related to the number of convolutional layers. If the number of convolutional layers is *N*, the receptive field size is (2*N* + 1) × (2*N* + 1). Therefore, the receptive field size in this study is 41 × 41. The activation function uses ReLu and uses zero padding to keep the input and output the same size. The complete spectrum measurement accuracy improvement process is shown in Figure 6. We averagely divide the measured skin spectral signal into 67 points in the visible light range (400~730 nm) to form 1 × 67 skin spectral data. Then the 1D skin spectral data of size 1 × 67 are stacked 40 times to form 2D information of size 41 × 67 and use this 2D information as input data of VDSR. Each row of the 2D information output by VDSR is a set of enhanced 1D skin spectral data. For stable and robust results, we averaged the middle 50% rows of the 2D information output by the VDSR to convert back to 1 × 67 1D skin spectral data.

In order to build an accurate spectrum enhancement model for VDSR, we need accurate spectral data, which is required for training. In this study, X-rite i1pro was selected as the reference spectral measuring instrument. The i1pro offers the advantages of a full-spectrum LED light source, accurate spectral measurements, and ease of use. The I1pro is hereinafter referred to as RS, which is the reference spectrometer. About the training spectral data, we invited 16 individuals to measure their skin spectrum at 9 parts, using the MSSMD and the RS, respectively. The 9 parts are the back of the left hand, the left arm, the back of the right hand, the right arm, the inner side of the left arm, the inner side of the right arm, the left cheek, the right cheek, and the forehead. Both devices collected 902 pieces of skin spectral data for training VDSR. Finally, the trained VDSR model is implanted into the system.

### 2.3. Mobile App

We used Android Studio and Java to design a mobile application to control the miniature skin spectrum measurement device, as shown in Figure 7. The mobile app displays the spectrum measurement results with CIE XYZ values and CIE xy chromaticity diagrams. A Bluetooth interface is used to connect the mobile phone and the MSSMD. The reason for choosing to design an app is that the mobile phone is the device most people own today. Moreover, using the app to design the control interface allows the user to control it more friendly and display the measurement results clearly. In addition, advanced skin-spectrum-related applications are added, including (1) EI/MI measurement, (2) Fitzpatrick-type classification, (3) Pantone SkinTone classification, (4) sun-exposure estimation, (5) body-fat measurement, etc. The calculation method is as follows:

#### 2.3.1. EI/MI Measurement

*EI* and *MI* can be calculated by (1) and (2), based on the research of Yoshimura et al. [20]:(1)$$EI=\frac{500}{\log5}\times \left(\log\left(\frac{red\;reflection\;880\;nm}{green\;reflection\;570\;nm}\right)+\log5\right)$$
(2)$$MI=\frac{500}{\log5}\times \left(\log\left(\frac{infrared\;reflection\;880\;nm}{red\;reflection\;660\;nm}\right)+\log5\right)$$

#### 2.3.2. Fitzpatrick-Type Classification

In terms of the Fitzpatrick type, according to the research of Bino and Bernerd [21], the human skin type can be calculated from the CIE Lab data of the subject’s skin tone using the individual typology angle (*ITA***°**) algorithm. The calculation method is shown in (3):(3)$$ITA^{{}^{\circ}}=\frac{\left(\frac{\arctan\left(L^{*}-50\right)}{b^{*}}\right)180}{3.14159}$$

The classification results are shown in Table 1.

#### 2.3.3. Pantone SkinTone Classification

CIELAB ΔE* is used to express the difference between two colors. The smaller the CIELAB ΔE*, the more similar the colors are. Therefore, we calculate the CIELAB ΔE* between the measured skin tone and each numbered color in the Pantone SkinTone Guide in the app. The number corresponding to the smallest CIELAB ΔE* is the Pantone SkinTone Guide number of the subject’s skin tone.

#### 2.3.4. Sun Exposure Estimation

According to the research of Webb et al. [22] and Terushkin et al. [23], the equations for the time required to synthesize vitamin D (*TRSVD*) and the recommended maximum sun exposure time (*RMSET*) are as follows, (4) and (5), where *SDD* is the standard vitamin D dose. *UV irradiance* is obtained from the current Bureau of Meteorology data. Depending on the type of skin tone, the energy required is shown in Table 2. The calculations are as follows:(4)$$TRSVD\left(s\right)=\frac{SDD\left(\frac{J}{m^{2}}\right)}{UV\;irradiance\left(\frac{W}{m^{2}}\right)}$$
(5)$$RMSET\left(s\right)=TRSVD\left(s\right)\times 4$$

#### 2.3.5. Body Fat Measurement

According to the patent of Robert D. Rosenthal [24], the body-fat percentage can be calculated by (6), where *l*_1_ is the reflected light intensity of 937 nm, *l*_2_ is the reflected light intensity of 947 nm, *W* is the weight (lbs), *H* is the height (inches), *S* is the gender (male is 0.01, and female is −0.01), and *EL* is the level of exercise (none is 0, mild is 0.02, moderate is 0.05, and severe is 0.08). *K*_0_ is an intercept error constant; *K*_2*A*_ and *K*_2*B*_ are the slopes of the curves, respectively, representing the two wavelengths being measured; and *K*_3_–*K*_6_ are for the respective body parameters. Moreover, *K*_0_*–K*_6_ are determined by a multiple regression technique. A set of example *K* values is *K*_0_ = 84.2, *K*_2*A*_ = −16.3, *K*_2*B*_ = −6.2, *K*_3_ = 8.1, *K*_4_ = −13.7, *K*_5_ = −124.2, and *K*_6_ = −81.4:(6)$$Body\;fat\left(\%\right)=K_{0}+K_{2A}\left(\log\frac{1}{l_{1}}\right)+K_{2B}\left(\log\frac{1}{l_{2}}\right)+K_{3}\left(\frac{W}{100}\right)+K_{4}\left(\frac{H}{100}\right)+K_{5}\left(S\right)+K_{6}\left(EL\right)$$

## 3. Developed System and Experimental Results

The LMSSMS developed in this study uses a low-cost MSSC NSP32, LED light source, 3D print housing, and control circuit to develop a low-cost MSSMD. The developed LMSSMD is shown in Figure 8a. The MSSMD contains an available rechargeable lithium battery and can also be powered by USB. We also developed an app to control MSSMD and showcase measurement results and skin-related mobile applications. The usage scenarios and mobile app are shown in Figure 8b. The indicator light of MSSMD starts to flash, which indicates that MSSMD has been initialized and is waiting for connection. At this point, we can use the app to control the connection between the mobile phone and MSSMD.

The app user interface is shown in Figure 9. First, we connect the app to MSSMD. After the connection button at the top of the screen is pressed, the nearby MSSMDs will be displayed according to their names, as shown in Figure 10a,b. Then we click on the desired device to make a connection. After the connection is completed, the connect button will turn into a disconnect button, as shown in Figure 10c. After the connection is complete, recalibration or loading of the calibration file is required, as shown in Figure 11a. After calibration, press Start to send the measurement command. The MSSMD starts the measurement after receiving the measurement command. The spectral data obtained by the MSSMD measurement will be sent back to the mobile phone. After the mobile phone receives the transmitted spectrum, it will display the waveform of the spectrum and its corresponding color. Moreover, the app performs spectrum analysis including EI/MI, Fitzpatrick skin-type classification, Pantone SkinTone, and body-fat percentage. If the mobile phone is connected to the Internet, the recommended sun-exposure hours can also be calculated according to the location (currently only in the Taiwan, Penghu, Kinmen, and Matsu area), as shown in Figure 11b.

To verify the accuracy of the developed LMSSMS, we compare the measurement results of the disabled-VDSR LMSSMS, the enabled-VDSR LMSSMS, and RS. The experiments are described as follows: The study invited 16 people to measure the spectrum of the skin, using the developed the disabled-VDSR LMSSMS, the enabled-VDSR LMSSMS, and RS, respectively. Each device collects 386 skin spectral data. Figure 12 is one of the spectral-comparison results of the disabled-VDSR LMSSMS, the enabled-VDSR LMSSMS, and RS. Table 3 is a comparison table of the average values of MSE. CIELAB ΔE* and Delta xy are calculated from the spectrum measured by the disabled-VDSR LMSSMS, the enabled-VDSR LMSSMS, and RS, respectively. In Figure 12, the LMSSMS (off) represents the spectrum measured by the disabled-VDSR LMSSMS. The LMSSMS (on) represents the spectrum measured by the enabled-VDSR LMSSMS. Standard represents the target spectrum measured by the RS. It can be observed that the LMSSMS (on) measurement is closer to the target spectrum measured by the RS than the LMSSMS (off). In Table 3, the MSE of the LMSSMS turned-off VDSR spectral signal is 7.99 10-4, CIELAB ΔE* = 3.28, and Δxy = 0.0097. The MSE of the LMSSMS turned-on VDSR enhanced spectral signal is only 2.06 10-4, CIELAB ΔE* = 1.95, and Δxy = 0.0046. The MSE of the LMSSMS turned-on VDSR and CIELAB ΔE* and Δxy are all smaller than LMSSMS turned-off VDSR. This means that VDSR can enhance the measurement accuracy. The CIELAB ΔE* < 2 and the Δxy < 0.005 between the spectra measured by LMSSMS with VDSR turned on and i1pro. All of the above confirm that the measurement results of the LMSSMS turned-on VDSR obtain good accuracy.

Table 4 shows a brief comparison of competitive products with the developed LMSSMS. Professional skin-tone or skincare measurement instruments such as CM-700d/CM-600d (Konica Minolta, Inc., Tokyo, Japan) [6] and Mexameter^®^ Mx 18 (Courage + Khazaka electronic GmbH, Cologne, Germany) [7] are highly accurate but very expensive. App-based skin-tone measurements such as Microskin™ ColourMatch (Kerim Health Ltd., Maidenhead, UK) [8] and My Best Colors (colorwise.me) [9] are easily accessible with a very low cost of ownership. However, the measurement accuracy can be affected by various factors, such as the quality of phone cameras, quality of white balance, and ambient light conditions, leading to biased results. Handheld skin-tone measurement devices such as PANTONE CAPSURE RM200-PT01 (X-Rite, Inc., Grand Rapids, MI, USA) [10] and SKINCOLORCATCH (Delfin, Kuopio, Finland) [11] have high measurement accuracy and various functions. It has high potential opportunities for daily use. The LMSSMS developed in this research combines low-cost MSSC, a simple optical design, and well-developed VDSR to achieve low-cost embracing great potentials for accurate measurement and enjoys the advantage that the computational cost is relatively much lower than the cost of optical components.

## 4. Conclusions

In this study, we developed and verified the proposed LMSSMS with DNN technology, which can implement accurate skin spectrum and achieve many applications, such as EI/MI measurement, Fitzpatrick skin-type classification, Pantone SkinTone classification, sun-exposure estimation, body-fat measurement, etc. In this study, the LMSSMS was divided into three parts: (1) MSSMD: The goal was to develop a low-cost, lightweight, portable skin spectrum measurement device, which is mainly divided into four parts: (i) Miniature spectrum measurement element: In this study, a NSP32 was innovatively used as the skin spectrum measurement element. The chip is currently the smallest chip with a low cost and can measure the spectrum in a wide wavelength range. (ii) Measurement light source: We used SunLike LEDs to provide the light source required for NSP32 measurements. The light source emitted by the LED can cover the spectral measurement range of the spectrum with uniform light intensity. (iii) Optical mechanism: In order to develop a compact and portable optical mechanism with NSP32 and SunLike LED as the core, a 3D design software Sketch Up and a 3D printer were used to create a simple optical mechanism. (iv) Control and communication interface: In order to process measurement data, receive measurement commands, and return measurement results, the microcontroller STM32F411 and Bluetooth chip nRF52832 were used to design the control circuit and communication interface of the MSSMD. (2) DNN: To achieve accurate skin spectrum measurements with a low-cost light source, low-cost sensor, and low-complexity optical mechanism, we proposed a DNN technique, VDSR, to improve measurement accuracy. (3) Mobile app: In order to allow the users to control and understand the measurement results clearly, a mobile app was designed. In addition to controlling and showcasing results, this app integrates advanced skin-spectrum-related applications, including the EI/MI measurement, Fitzpatrick skin-type classification, Pantone SkinTone classification, sun-exposure estimation, body-fat measurement, etc. Finally, the skin spectra collected by the developed LMSSMS were compared with standard instruments. The comparison results show that the measurement accuracy of the LMSSMS can reach 94.7%. This result has verified the feasibility of developing a miniature and reliable skin spectroscopy measurement system that uses a low-cost micro-spectroscopy chip combined with DNN technology. In this study, the experimental results show that the currently used DNN technology, VDSR, performs quite well. However, in the future, we can further explore what kind of DNN technology can make the measurement results of the LMSSMS more accurate.

## Figures and Tables

**Figure 1 sensors-22-08844-f001:**
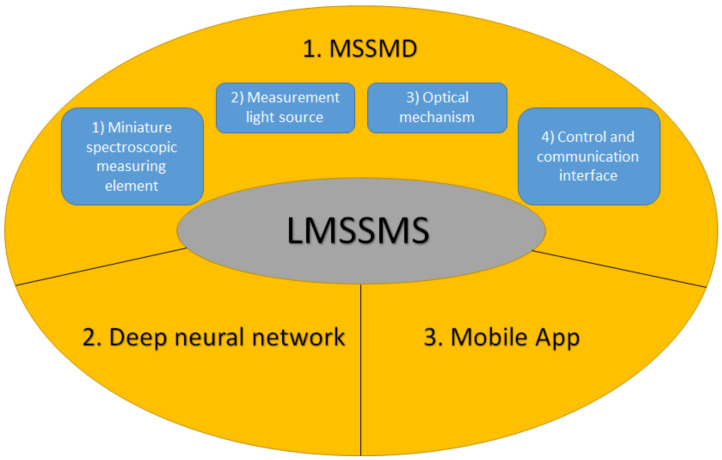
The structure of the developed LMSSMS.

**Figure 2 sensors-22-08844-f002:**
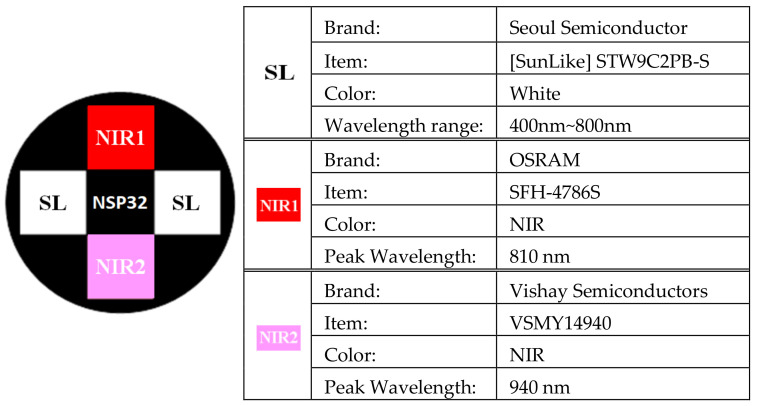
The light source configurations.

**Figure 3 sensors-22-08844-f003:**
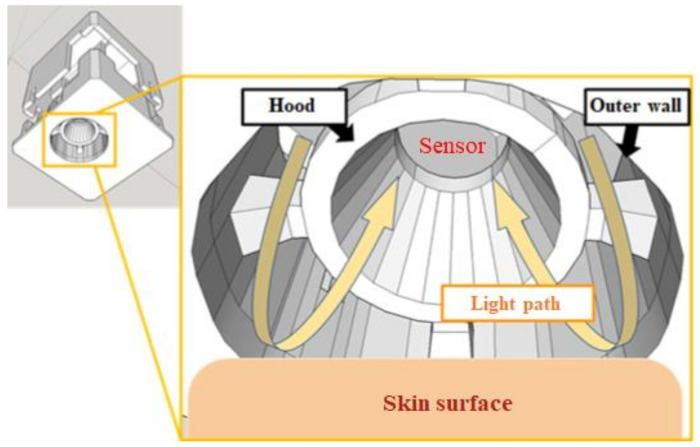
The optical-path design concept.

**Figure 4 sensors-22-08844-f004:**
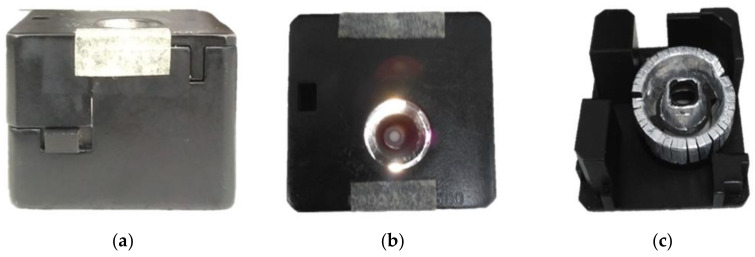
Photos of the finished device housing: (**a**) side view, (**b**) top view, and (**c**) optical path channel.

**Figure 5 sensors-22-08844-f005:**
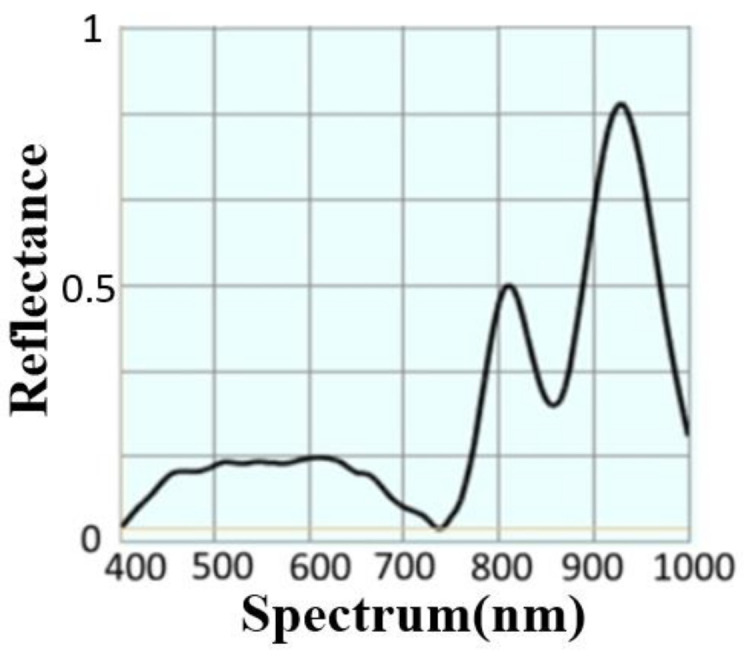
The reflectance spectrum of MSSMD measured on the diffuse reflectance standard whiteboard WS-1-SL.

**Figure 6 sensors-22-08844-f006:**
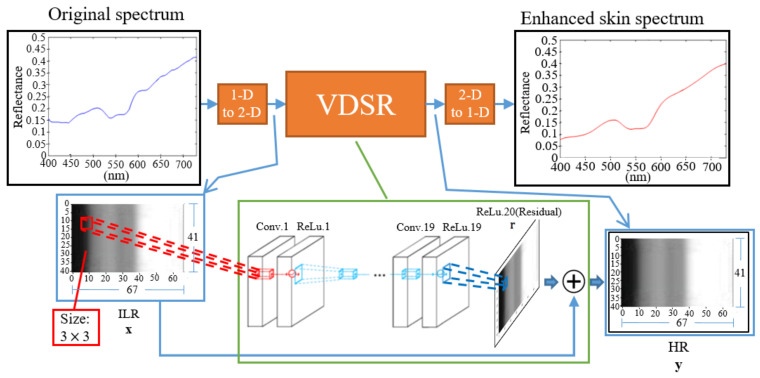
Proposed VDSR network architecture.

**Figure 7 sensors-22-08844-f007:**
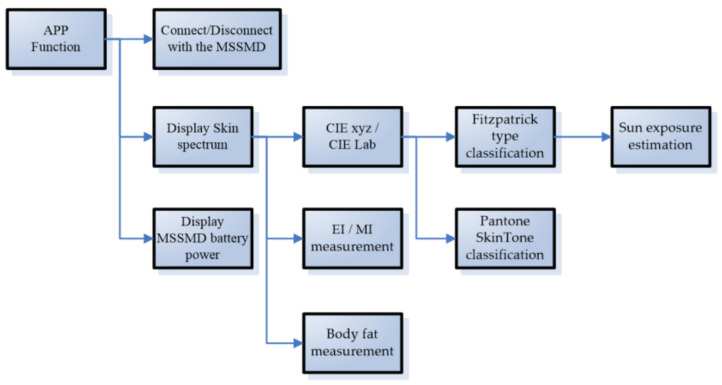
Mobile application functions.

**Figure 8 sensors-22-08844-f008:**
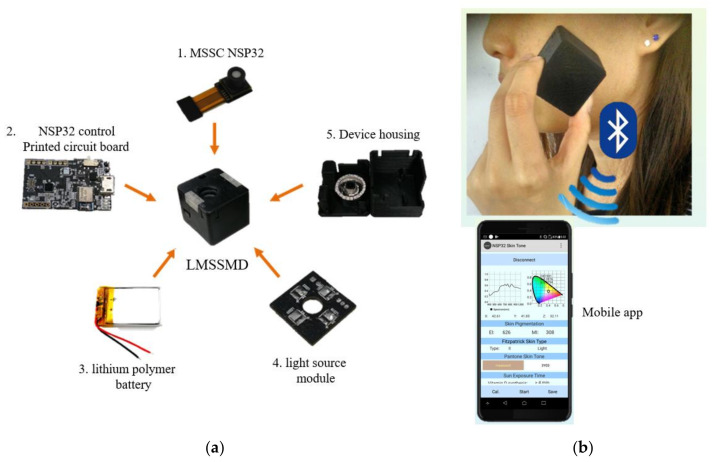
The LMSSMS: (**a**) LMSSMD and (**b**) usage scenarios and mobile app.

**Figure 9 sensors-22-08844-f009:**
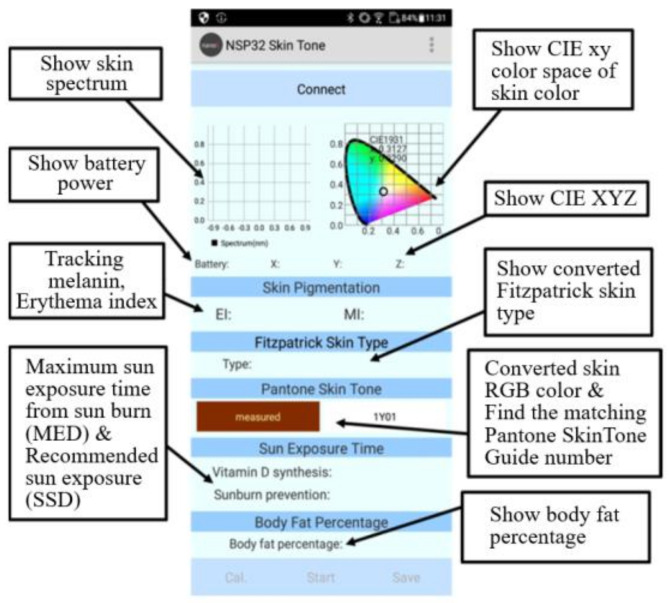
The app user interface.

**Figure 10 sensors-22-08844-f010:**
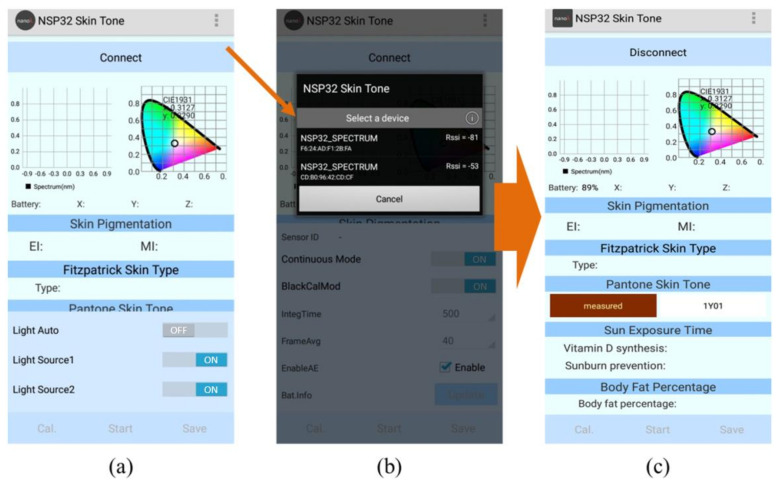
The operation process of app connection: (**a**) UI initial, (**b**) connection device, and (**c**) device connected and ready to use.

**Figure 11 sensors-22-08844-f011:**
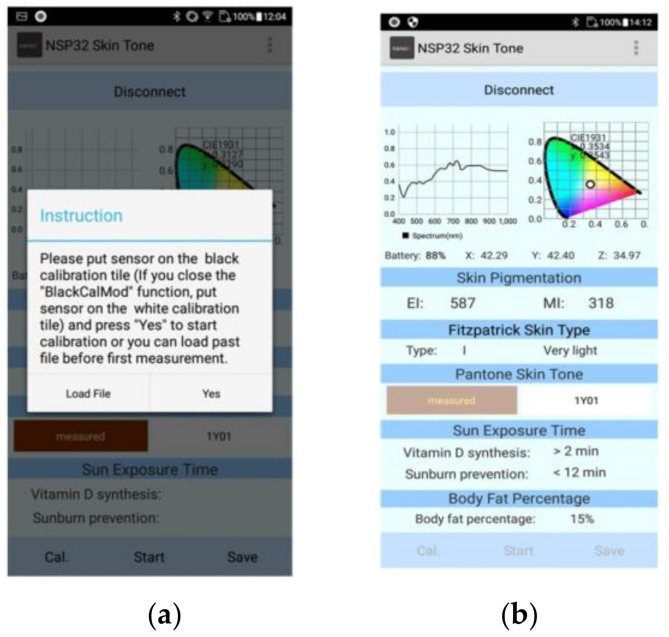
App measurement operation and measurement results: (**a**) recalibration or loading of the calibration file; (**b**) skin spectral analysis and related application results.

**Figure 12 sensors-22-08844-f012:**
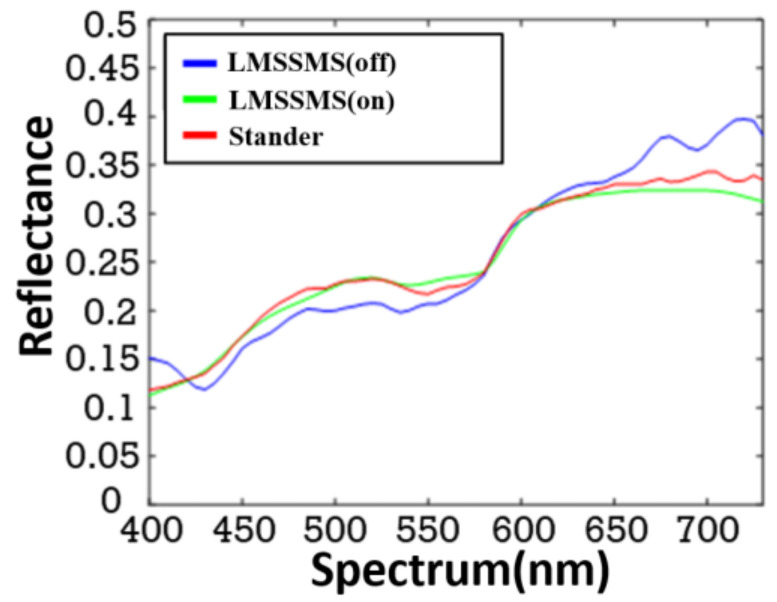
The LMSSMS turned-off VDSR measurement spectrum vs. LMSSMD turned-on VDSR vs. reference spectrometer i1pro.

**Table 1 sensors-22-08844-t001:** Fitzpatrick skin-type classification results [21].

Individual Typology Angle (*ITA*°)	Skin Classification
*ITA*° > 55°	Very light
41° < *ITA*° < 55°	Light
28° < *ITA*° < 41°	Intermediate
10° < *ITA*° < 28°	Tan
−30° < *ITA*° < 10°	Brown
*ITA*° < −30°	Dark

**Table 2 sensors-22-08844-t002:** Standard vitamin D dose [22].

Skin Type	Color	1 SSD (J m-2D Effective)
Type I	Caucasian; blonde or red hair, freckles, fair skin, blue eyes	37.2
Type II	Caucasian; blonde or red hair, freckles, fair skin, blue eyes or green eyes	46.5
Type III	Darker Caucasian, light Asian	55.8
Type IV	Mediterranean, Asian, Hispanic	83.6
Type V	Middle Eastern, Latin, light-skinned black, Indian	111.4
Type VI	Dark-skinned black	185.1

**Table 3 sensors-22-08844-t003:** Comparisons of MSE, CIELAB ΔE*, and Delta xy performance of LMSSMS turned-off/on VDSR measurement spectrum vs. reference spectrometer i1pro.

Comparing Results	LMSSMS Turned-Off VDSR Measurement Spectrum vs. Reference Spectrometer i1pro	LMSSMS Turned-On VDSR Measurement Spectrum vs. Reference Spectrometer i1pro
MSE	7.99 × 10^−4^	2.06 × 10^−4^
CIELAB ΔE*	3.28	1.95
Δxy	0.0097	0.0046

**Table 4 sensors-22-08844-t004:** Comparison between LMSSMS and market available products.

	Measuring System	Measuring Element	Light Path Design	AI Enhancement	Accuracy	Cost	Providing Functions
Our Developed	LMSSMS	Miniature spectrum sensing chip	Simple design	O(Spectral enhancement)	High	Low	Skin-tone measurement, EI/MI measurement, Fitzpatrick skin-type classification, Pantone SkinTone classification, sun-exposure estimation, and body-fat measurement
Professional	Konica Minolta CM-700d/CM-600d [6]	Professional spectrometry components	Dedicated Design	X	Very high	Very high	Skin-tone measurement
	Courage+Khazaka Mexameter^®^ Mx 18 [7]	Professional spectrometry components	Dedicated Design	X	Very high	Very high	EI/MI measurement
Handheld	X-Rite PANTONE CAPSURE RM200-PT01 [8]	Professional spectrometry components	Dedicated Design	X	High	High	Skin-tone measurement and Pantone SkinTone classification
	Delfin SKINCOLORCATCH [9]	Professional spectrometry components	Dedicated Design	X	High	High	Skin-tone measurement, EI/MI measurement, and Fitzpatrick skin-type classification
App	Microskin™ ColourMatch [10]	Phone or tablet camera	None (Phone or tablet lens)	X	Low	Very low	Skin-tone measurement
	My Best Colors [11]	Phone or tablet camera	None (Phone or tablet lens)	X	Low	Very low	Skin-tone measurement and makeup recommendations

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
