# Peer review of "Developing Low-Cost Mobile Device and Apps for Accurate Skin Spectrum Measurement via Low-Cost Spectrum Sensors and Deep Neural Network Technology"

_sensors, 2022, doi:10.3390/s22228844_

Round 1
Reviewer 1 Report
This paper presents an integrated portable device for skin spectrum measurement as well as operation and analysis software. By using The whole system worked well for EI/MI measurement, Fitzpatrick type classification, Pantone SkinTone classification, Sun exposure estimation, and body fat measurement, etc. at good accuracy.
The topic is important. The results are reliable and valuable to researchers in related fields. The paper is well written. Only minor revision is needed to correct typos and grammar mistakes.
Author Response
Thanks for your valuable comments.
Point 1:
This paper presents an integrated portable device for skin spectrum measurement as well as operation and analysis software. By using The whole system worked well for EI/MI measurement, Fitzpatrick type classification, Pantone SkinTone classification, Sun exposure estimation, and body fat measurement, etc. at good accuracy.
The topic is important. The results are reliable and valuable to researchers in related fields. The paper is well written. Only minor revision is needed to correct typos and grammar mistakes.
Response 1:
We already have St. Paul Workshop (http://www.ahkue.com/ : a professional editing and proofreading Workshop) to revise typos and grammatical errors.
Reviewer 2 Report
This paper reports a new mobile device for human skin spectrum measurement using a new algorithm. The results are solid and sound, and manuscript writing is clear. I recommend the acceptance of this paper with minor revision: the authors should add a comparison of their device with other state-of-the-art devices to show the accuracy and cost advantages.
Author Response
Thanks for your valuable comments.
Point 1:
This paper reports a new mobile device for human skin spectrum measurement using a new algorithm. The results are solid and sound, and manuscript writing is clear. I recommend the acceptance of this paper with minor revision: the authors should add a comparison of their device with other state-of-the-art devices to show the accuracy and cost advantages.
Response 1:
We have updated the comparison with other techniques or instruments in lines 46-56 of section 1, in lines 294-307 of section 3, and added a comparison table in Table 4.
Content in lines 46-56 of section 1:
However, currently available skin tone or skin spectrum measuring devices present different problems for individual daily use. Although professional instruments such as Konica Minolta CM-700d / CM-600d [6], and Courage+Khazaka Mexameter® Mx 18 [7] can serve as golden standard, they are bulky, heavy, and expensive. Apps that use smartphone's built-in cameras, such as Microskin™ ColourMatch [8] and My Best Colors [9] are easy to use, but are likely to be affected by many factors such as the camera quality, shooting environments, and white balance calibration, leading to inaccurate results. On the other hand, hand-held skin tone measurement devices such as X-Rite PANTONE CAPSURE RM200-PT01 [10] and Delfin SKINCOLORCATCH [11] have emerged on the market and showed a great potential for daily use applications, due to a reasonable price, accuracy and easy usage.
Content in lines 294-307 of section 3:
Table 4 shows a brief comparison of competitive products with the developed LMSSMS. Professional skintone or skincare measurement instrument such as Konica Minolta CM-700d / CM-600d [6] and Courage+Khazaka Mexameter® Mx 18 [7] are highly accurate but very expensive. APP-based skintone measurement such as Microskin™ ColourMatch [8] and My Best Colors [9] are easily accessible with a very low cost of owner-ship. However, the measurement accuracy can be affected by various factors such as the quality of phone cameras, quality of white balance, and ambient light conditions, leading to biased results. Hand-held skintone measurement device such as X-Rite PANTONE CAPSURE RM200-PT01 [10] and Delfin SKINCOLORCATCH [11] have high measurement accuracy and various functions. It has highly potential opportunities for daily use. The LMSSMS developed in this research combines low-cost MSSC, a simple optical design, and well-developed VDSR to achieve low cost embracing great potentials for accurate measurement, and enjoys the advantage that the computational cost is relatively much lower than the cost of optical components.
Table 4. Comparison among LMSSMS and market available products.
|
Measuring System |
Measuring element |
Light Path Design |
AI Enhancement |
Accuracy |
Cost |
Providing Functions |
Our Developed |
LMSSMS |
Miniature spectrum sensing chip |
Simple design |
O(Spectral enhancement) |
High |
Low |
Skin tone measurement, EI/MI measurement, Fitzpatrick skin type classification, Pantone SkinTone classification, Sun exposure estimation, and Body fat measurement |
Professional |
Konica Minolta CM-700d / CM-600d [6] |
Professional spectrometry components |
Dedicated Design |
X |
Very high |
Very high |
Skin tone measurement |
Courage+Khazaka Mexameter® Mx 18 [7] |
Professional spectrometry components |
Dedicated Design |
X |
Very high |
Very high |
EI/MI measurement |
|
Handheld |
X-Rite PANTONE CAPSURE RM200-PT01 [8] |
Professional spectrometry components |
Dedicated Design |
X |
High |
High |
Skin tone measurement, and Pantone SkinTone classification |
Delfin SKINCOLORCATCH [9] |
Professional spectrometry components |
Dedicated Design |
X |
High |
High |
Skin tone measurement, EI/MI measurement, and Fitzpatrick skin type classification |
|
APP |
Microskin™ ColourMatch [10] |
Phone or tablet camera |
None (Phone or tablet lens ) |
X |
Low |
Very low |
Skin tone measurement |
My Best Colors [11] |
Phone or tablet camera |
None (Phone or tablet lens ) |
X |
Low |
Very low |
Skin tone measurement, and makeups recommendation |
Reviewer 3 Report
This paper presents the development of a skin spectroscopy measurement device that has a deep learning backbone aimed at enhancing the device's accuracy.
I think the literature is rather poor and the paper does not clearlyposition the developments with respect to other similar devices or technologies.
I would suggest that you add some more information to your background section and also cover a very important aspect of deep learning, i.e. uncertainty estimation:
De Sousa Ribeiro, F., Calivá, F., Swainson, M., Gudmundsson, K., Leontidis, G. and Kollias, S., 2020. Deep bayesian self-training. Neural Computing and Applications, 32(9), pp.4275-4291.
Lastly, in the abstract, rather than saying "In this paper, based on the experience of using convolutional neural network (CNN) to enhance the classification accuracy of Fitzpatrick skin type in 2018", you should specify that this refers to Fitzpatrick's developed method proposed in 2018, as it is not clear what you are referring to here.
Author Response
Thanks for your valuable comments.
Point 1:
This paper presents the development of a skin spectroscopy measurement device that has a deep learning backbone aimed at enhancing the device's accuracy.
I think the literature is rather poor and the paper does not clearlyposition the developments with respect to other similar devices or technologies.
Response 1:
We have updated the comparison with other techniques or instruments in lines 46-56 of section 1, in lines 294-307 of section 3, and added a comparison table in Table 4.
Content in lines 46-56 of section 1:
However, currently available skin tone or skin spectrum measuring devices present different problems for individual daily use. Although professional instruments such as Konica Minolta CM-700d / CM-600d [6], and Courage+Khazaka Mexameter® Mx 18 [7] can serve as golden standard, they are bulky, heavy, and expensive. Apps that use smartphone's built-in cameras, such as Microskin™ ColourMatch [8] and My Best Colors [9] are easy to use, but are likely to be affected by many factors such as the camera quality, shooting environments, and white balance calibration, leading to inaccurate results. On the other hand, hand-held skin tone measurement devices such as X-Rite PANTONE CAPSURE RM200-PT01 [10] and Delfin SKINCOLORCATCH [11] have emerged on the market and showed a great potential for daily use applications, due to a reasonable price, accuracy and easy usage.
Content in lines 294-307 of section 3:
Table 4 shows a brief comparison of competitive products with the developed LMSSMS. Professional skintone or skincare measurement instrument such as Konica Minolta CM-700d / CM-600d [6] and Courage+Khazaka Mexameter® Mx 18 [7] are highly accurate but very expensive. APP-based skintone measurement such as Microskin™ ColourMatch [8] and My Best Colors [9] are easily accessible with a very low cost of owner-ship. However, the measurement accuracy can be affected by various factors such as the quality of phone cameras, quality of white balance, and ambient light conditions, leading to biased results. Hand-held skintone measurement device such as X-Rite PANTONE CAPSURE RM200-PT01 [10] and Delfin SKINCOLORCATCH [11] have high measurement accuracy and various functions. It has highly potential opportunities for daily use. The LMSSMS developed in this research combines low-cost MSSC, a simple optical design, and well-developed VDSR to achieve low cost embracing great potentials for accurate measurement, and enjoys the advantage that the computational cost is relatively much lower than the cost of optical components.
Table 4. Comparison among LMSSMS and market available products.
|
Measuring System |
Measuring element |
Light Path Design |
AI Enhancement |
Accuracy |
Cost |
Providing Functions |
Our Developed |
LMSSMS |
Miniature spectrum sensing chip |
Simple design |
O(Spectral enhancement) |
High |
Low |
Skin tone measurement, EI/MI measurement, Fitzpatrick skin type classification, Pantone SkinTone classification, Sun exposure estimation, and Body fat measurement |
Professional |
Konica Minolta CM-700d / CM-600d [6] |
Professional spectrometry components |
Dedicated Design |
X |
Very high |
Very high |
Skin tone measurement |
Courage+Khazaka Mexameter® Mx 18 [7] |
Professional spectrometry components |
Dedicated Design |
X |
Very high |
Very high |
EI/MI measurement |
|
Handheld |
X-Rite PANTONE CAPSURE RM200-PT01 [8] |
Professional spectrometry components |
Dedicated Design |
X |
High |
High |
Skin tone measurement, and Pantone SkinTone classification |
Delfin SKINCOLORCATCH [9] |
Professional spectrometry components |
Dedicated Design |
X |
High |
High |
Skin tone measurement, EI/MI measurement, and Fitzpatrick skin type classification |
|
APP |
Microskin™ ColourMatch [10] |
Phone or tablet camera |
None (Phone or tablet lens ) |
X |
Low |
Very low |
Skin tone measurement |
My Best Colors [11] |
Phone or tablet camera |
None (Phone or tablet lens ) |
X |
Low |
Very low |
Skin tone measurement, and makeups recommendation |
Point 2:
I would suggest that you add some more information to your background section and also cover a very important aspect of deep learning, i.e. uncertainty estimation:
De Sousa Ribeiro, F., Calivá, F., Swainson, M., Gudmundsson, K., Leontidis, G. and Kollias, S., 2020. Deep bayesian self-training. Neural Computing and Applications, 32(9), pp.4275-4291.
Response 2:
The recommended nature paper is added and commented in lines 66-76 in section 1.
Content in lines 66-76 of section 1:
Supervised deep learning has been highly successful on classification in recent years, achieving state-of-the-art results in many tasks. We conducted Fitzpatrick skin type classification with a self-developed simple device using convolutional neural network (CNN) in 2018 [14], achieving a 92.59% classification accuracy. Recently, deep learning classification has been developed in the era of self-annotation, active sampling, and self-learning, making the classification tasks even more intelligent [15]. Apart from classification, this study is trying on super resolution capability of deep learning. For proof-of-concept development, this study adopted the well-developed VDSR for spectral enhancement. Leveraging on the developing experience in [14], this study developed a low-cost miniature skin spectrum measurement system (LMSSMS) using deep neural network (DNN) technology.
Point 3:
Lastly, in the abstract, rather than saying "In this paper, based on the experience of using convolutional neural network (CNN) to enhance the classification accuracy of Fitzpatrick skin type in 2018", you should specify that this refers to Fitzpatrick's developed method proposed in 2018, as it is not clear what you are referring to here.
Response 3:
We are sorry for causing you confusion. We have updated the Abstract in lines 12-16.
Content in lines 12-16 of Abstract:
We used convolutional neural network (CNN) to achieve a satisfying accuracy of Fitzpatrick skin type classification using a self-developed simple device in 2018. Leveraging on the hardware, firmware and software APP developing experience, a low-cost miniature skin spectrum measurement system (LMSSMS) using deep neural network (DNN) technology is further studied and the feasibility of the system is verified in this paper.